# Targeted Delivery of Abaloparatide to Spinal Fusion Site Accelerates Fusion Process in Rats

**DOI:** 10.3390/biomedicines12030612

**Published:** 2024-03-08

**Authors:** Jeffery J. Nielsen, Stewart A. Low, Christopher Chen, Xinlan Li, Ephraim Mbachu, Lina Trigg, Siyuan Sun, Madeline Tremby, Rahul Hadap, Philip S. Low

**Affiliations:** 1Department of Medicinal Chemistry and Molecular Pharmacology, Purdue University, West Lafayette, IN 47907, USA; nielsej@purdue.edu; 2Novosteo Inc., West Lafayette, IN 47906, USA; 3Department of Chemistry, Purdue University, West Lafayette, IN 47907, USA

**Keywords:** spinal fusion repair, targeted drug delivery, hydroxyapatite targeting, osteoanabolic, abaloparatide, acidic oligopeptides

## Abstract

Spinal fusions are performed to treat congenital skeletal malformations, spondylosis, degenerative disk diseases, and other pathologies of the vertebrae that can be resolved by reducing motion between neighboring vertebrae. Unfortunately, up to 100,000 fusion procedures fail per year in the United States, suggesting that efforts to develop new approaches to improve spinal fusions are justified. We have explored whether the use of an osteotropic oligopeptide to target an attached bone anabolic agent to the fusion site might be exploited to both accelerate the mineralization process and improve the overall success rate of spinal fusions. The data presented below demonstrate that subcutaneous administration of a modified abaloparatide conjugated to 20 mer of D-glutamic acid not only localizes at the spinal fusion site but also outperforms the standard of care (topically applied BMP2) in both speed of mineralization (*p* < 0.05) and overall fusion success rate (*p* < 0.05) in a posterior lateral spinal fusion model in male and female rats, with no accompanying ectopic mineralization. Because the bone-localizing conjugate can be administered ad libitum post-surgery, and since the procedure appears to improve on standard of care, we conclude that administration of a bone-homing anabolic agent for improvement of spinal fusion surgeries warrants further exploration.

## 1. Introduction

Over 500,000 spinal fusions are performed annually in the United States at an average cost of ~USD 120,000/per procedure [1]. With recovery times for full activity ranging from 3 to 12 months [2] and failure rates varying from 10 to 36% [3,4], there is clearly a need to explore alternative strategies for improving the procedure [1]. Thus, prolonged postoperative immobilization increases the risk for comorbidities and reduces quality of life, while fusion failure imposes a need for additional surgical interventions and/or prolonged restriction of mobility [5]. Moreover, current therapeutic approaches rely on the intraoperative local application of bone morphogenetic protein (BMP), which can leak from the fusion site and cause ectopic mineralization, osteoclast-mediated bone resorption, and inappropriate adipogenesis [1,6,7,8]. Moreover, in many cases, the spinal fusion is temporarily stabilized with a metal rod that can loosen before fusion is complete, leading to a requirement for additional surgical procedures [3,9]. Many advances have been made in terms of improving devices to improve therapeutic outcomes [10,11,12,13]. However, advances in terms of utilizing the underlying biology to improve the repair of a spinal fusion have been relatively limited [14,15,16]. This limitation has partly been due to the challenges of selectively promoting repair of the spinal fusion over the rest of the skeleton, which requires spatiotemporal control over the therapeutic [14]. Attempts to address this issue have been made with local applications of therapeutics, but this method struggles to maintain enough of a therapeutic window [15,17].

To improve clinical outcomes for patients who receive spinal fusion surgery we aimed to develop a noninvasively applied therapeutic to augment current clinical practices and accelerate and improve the healing of spinal fusions. To address some of the current spinal fusion clinical practice problems, our lab developed a targeting platform that can target a bone anabolic agent specifically to fusing vertebrae following systemic injection. This platform can thereby noninvasively concentrate the osteogenic activity at the fusion site and restrict off-target bone generation and allow for repeat dosing, which enables longer durations of therapeutic effect. We have previously demonstrated this targeting platform, which is based on certain acidic peptides that will localize to exposed hydroxyapatite at the site of a bone fracture following systemic injection. We have also demonstrated that it can be used to deliver tethered bone anabolic to the exposed bone surface [18,19,20,21,22,23]. In the case of spinal fusions, we hypothesized that this targeting system could be used to deliver abaloparatide, a known bone anabolic agent, to the decorticated vertebral surfaces created during the spinal fusion surgery to accelerate repair and improve the success rate of the fusion. In this paper, we confirm this hypothesis in a rat posterolateral lumbar spinal fusion model [24,25,26] by demonstrating that our hydroxyapatite-homing peptide will concentrate attached drugs specifically on fusion sites containing scaffolds composed of collagen sponges, bone granules, or mineralized collagen to structurally guide the fusion process. Because this localization of abaloparatide at the fusion site is further shown to accelerate the fusion process while also increasing the fraction of stable unions, we conclude that the use of a systemically administered hydroxyapatite targeting peptide that can accumulate at the fusion site can be exploited to continuously stimulate bone regeneration during the healing process, leading to a more rapid and complete fusion outcome.

## 2. Methods

### 2.1. PTH1R Signaling Assay

The activity of Ab46_D-Glu20 though PTH1R was evaluated in UMR-106 cells (ATCC, Manassas, VA, USA) which express PTH1R. Cells were plated in a 96-well plate at 15,000 cells per well. After 24 h, the cells were treated with serial dilutions of abaloparatide or Ab_46_-D Glu_20_ at 37 °C and 5% CO_2_ for 1 h. After the 1 h induction, the activity of the PTH1R was evaluated by quantifying cAMP production with Promega’s cAMP-Glo™ Assay (Promega, Madison, WI, USA) according to their published procedure. Luminescence was collected for 1 s. Each sample was run in triplicate along with an ATP standard curve, and evaluated relative to untreated controls. Data were analyzed using GraphPad Prisms version 9 software, La Jolla, CA, USA.

### 2.2. Hydroxyapatite Binding Study

To evaluate the binding of Ab_46_-D Glu_20_ to hydroxyapatite, an ELISA was performed with magnetic beads conjugated to hydroxyapatite (Bioclone Inc., San Diego, CA, USA). The targeting portion of Ab_46_-D Glu_20_ is designed to bind to hydroxyapatite and is captured by these beads. A constant quantity of hydroxyapatite-conjugated magnetic beads was seeded in each of the wells. The wells were then treated with increasing concentrations of Ab_46_-D Glu_20_, the bound complex was pulled down using a magnetic block, and the unbound peptide was washed off. Then, a mouse monoclonal Ab that was raised against a consensus sequence (CPTC-PTHrP-1) between Ab_46_-D Glu_20_ and PTHrP was used as the primary Ab (University of Iowa Developmental Studies Hybridoma Bank, Iowa City, IA, USA). Following incubation with an HRP-conjugated anti-mouse secondary Ab (Promega, Madison, WI, USA) and subsequent treatment with a luminescent HRP substrate MB Chemiluminescence ELISA substrate (Roche, Basel, Switzerland), luminescence was read using a plate reader, and the values were plotted in GraphPad Prism version 9 as a function of peptide concentration to generate the binding curves and determine the Kd. The Kd of Ab_46_-D Glu_20_ to be 26.5 × 10^−9^ M.

### 2.3. Biodistribution Study

To evaluate the ability to localize therapeutics to spinal fusions, female Sprague Dawley rats (Envigo, Indianapolis, IN, USA) (n = 3) underwent a L4–L5 posterolateral lumbar spinal fusion using collagen sponge (Ace Surgical Supply, Brockton, MA, USA) and inorganic bone granules wrapped in collagen or a mineralized collagen scaffold soaked in 10 μg BMP2 (BioVision, Milpitas, CA, USA). The bilateral posterolateral lumbar spinal fusion was performed on rats under aseptic conditions. The muscle on the vertebral body between the spinous process and the transverse process was bluntly dissected away from the vertebral body to expose the transverse processes. Using a low-speed bur, the L4 and L5 transverse processes’ cortical bone and the facet joint between the two vertebrae were removed, and a drug-soaked collagen sponge was placed on each side of the vertebrae. The scaffold was soaked to saturation in the drugs for 10 min. It was imaged in a μCT (Perkin Elmer, Waltham, MA, USA) longitudinally every week for 8 weeks and injected with a near-infrared dye (S0456) acidic oligopeptide conjugate and imaged 24 h later with a spectral imaging system, AMI (Spectral imaging instruments, Tucson, AZ, USA) for 1 s with excitation at 745 nM and emission collected at 810 nM.

### 2.4. Efficacy Study

Once acidic oligopeptides spinal fusion-targeting was established, an efficacy study in both male and female skeletally-mature Sprague Dawley rats (n = 10/sex) (Envigo, Indianapolis, IN, USA) was performed using the same surgical procedure as before with a 5 × 7.5 mm RCM6 collagen sponge (Ace Surgical supplies, Brockton, MA, USA). A total of 60 rats were randomly assigned to one of three treatments: phosphate-buffered saline vehicle control, 10 μg BMP2 (BioVision, Milpitas, CA, USA), or 38 nmol/kg twice a week of acidic oligopeptide-targeted abaloparatide (Abalo46-D-Glu20). The initial dose of all drugs was administered on the sponge and then vehicle control and Abalo46-D-Glu20 were administered subcutaneously twice a week for 8 weeks. The spinal fusion mineralization was assessed using a Quantum GX μCT (Perkin Elmer, Waltham, MA, USA). Each rat was scanned weekly during weeks 3–8. The scans were blindly scored by two separate individuals. The density and total volume of the fusions was calculated from the μCT scans. At the end, the lumbars were removed and blindly manually palpated, as has been previously described [27,28]. Data were analyzed in GraphPad Prism version 9, and significance was established by ANOVA. All animal experiments were performed in accordance with protocols approved by Purdue University’s Institutional Animal Care and Use Committee (IACUC).

## 3. Results

To create a hydroxyapatite-targeted bone anabolic agent, we synthesized a variant of abaloparatide and linked it to a linear 20 mer of D-glutamic acid that naturally accumulates on exposed hydroxyapatite (Figure 1, [20]). In order to assure that this conjugate can stimulate parathyroid hormone receptor 1 (PTH1R) with the same potency as unmodified abaloparatide, we compared the abilities of our conjugate and unmodified abaloparatide to stimulate cAMP production in PTH1R-expressing UMR106 cells. As shown in Figure 2A, the targeted conjugate (Ab_46_-D Glu_20_) was found to induce cAMP biosynthesis with a similar affinity to unmodified abaloparatide (EC_50_~0.5 nM), (i.e., suggesting that the C-terminal oligopeptide extension does not interfere with abaloparatide’s ability to signal through PTH1R [29,30,31]). Additionally, we found that the acidic oligopeptide retained its high affinity for hydroxyapatite after conjugation to the abaloparatide, yielding a K_D_~27 nM (Figure 2B).

We next evaluated the ability of the hydroxyapatite targeting peptide (D-Glu_20_) to concentrate a fluorescent reporter molecule (S0456) at the spinal fusion site in a rat L4–L5 posterolateral lumbar spinal fusion model [8,32,33,34,35]. For this purpose, we performed spinal fusion surgeries using the three most common forms of graft material, namely, collagen sponges, mineralized sponges, and bone granules (see Figure 3B), each containing rh-BMP2 for stimulation of bone regeneration [34]. As shown in Figure 3A, subcutaneous injection of the fluorescent D-Glu_20_ conjugate resulted in localization of the fluorescence almost exclusively at the fusion site, with some fluorescent conjugate appearing on the feet and/or heads of the injected rats. Because dissection of the rats revealed that the fluorescence observed on the feet and heads was localized to the fur, we speculate that this dye was simply excreted in the urine and transferred to the fur during grooming rather than delivered to the hair via the vasculature. It was also encouraging to note that all three scaffold materials were capable of capturing the fluorescent oligopeptide, suggesting that all three scaffolds should also accumulate Ab_46_-D Glu_20_ following its systemic injection.

With the acidic oligopeptide’s targeting ability established, we next elected to determine whether the similarly targeted bone anabolic agent, Ab_46_-D Glu_20_, might improve the fusion process in skeletally mature Sprague Dawley rats. For this purpose, both male and female rats (*n* = 10/group/sex) were subjected to an L4–L5 bilateral posterolateral lumbar spinal fusion procedure in which a collagen sponge (RCM6 Ace Surgical Supplies) was implanted as the structural scaffold. Rats of each sex were then randomly assigned to one of three treatment groups (phosphate-buffered saline vehicle control, 10 μg BMP2, or 38 nmol/kg b.i.w. Ab_46_-D Glu_20_), and an initial dose of each treatment was applied to the installed sponge. Thereafter, vehicle control and Ab_46_-D-Glu_20_ were administered subcutaneously twice a week for 8 weeks, and spinal mineralization was assessed weekly by in vivo μCT. Scans were blindly scored by two individuals, and bone density and total volumes of the fusions were calculated. As shown in Figure 4A, analysis of the μCT images from the rat with the median blinded fusion score in each treatment group (*n* = 10) revealed greater mineralization at week 5 post-surgery in the Ab_46_-D Glu_20_ than in the other two cohorts (comparing mineralization of the gaps between the vertebrae among cohorts). Moreover, analysis of the changes in μCT images as a function of time demonstrated that mineralization also occurred faster in the Ab_46_-D Glu_20_ group than either the BMP- or saline-treated cohorts. Although this rapid mineralization was seen in both male and female rats, female rats responded more poorly to BMP2 than they did to Ab_46_-D-Glu_20_, which is an established issue with BMP2 [27]. Due to this effect, the remainder of our studies were focused on male rats (Figure 5, Figure 6 and Figure 7).

Figure 6 shows the average fusion scores for each treatment group (*n* = 10/group) as a function of time after surgery. To obtain these average scores, each rat was evaluated by two blinded reviewers to assign a score of 0 to 4 for osteointegration, fusion bone density, bridging on the left side, and bridging on the right side of the spine, for a maximum possible score of 16. As revealed by the average fusion scores in Figure 6, Ab_46_-D-Glu_20_ was found to induce fusion more effectively and rapidly than either BMP or saline. Indeed, by the 8-week time point, the Ab_46_-D-Glu_20_ group had achieved a fusion rate of 93% compared to 63% for the BMP2-treated group and 38% in the saline group (Figure 7A). Not surprisingly, mechanical bending (manual palpitation) of the L4–L5 joint [27,28] yielded a similar conclusion, in that 82% of the Ab_46_-D-Glu_20_-treated joints were confirmed to have fused in contrast to only 66% and 29% of the BMP2- and saline-treated joints, respectively (Figure 7B). Perhaps most importantly, the Ab_46_-D-Glu_20_ group reached an average fusion score of 14.3 out of a possible 16 by the 8-week time point, whereas the BMP2 and saline groups attained scores of only 9.2 and 6.2 by the same endpoint (Figure 6). Whether these latter two groups would have eventually achieved a completely healed score of 16 was not determined, but based on the trajectory of the Ab_46_-D-Glu_20_ group, it can be speculated that the rats in this group would have eventually experienced a complete repair. Taken together, these data argue that Ab_46_-D-Glu_20_ significantly accelerates the fusion process without compromising the quality of the final product.

## 4. Discussion

Recent studies have shown that acidic peptides can target attached cargoes specifically to exposed hydroxyapatite on bone fracture surfaces, thereby enabling accelerated fracture healing by concentrating bone anabolic agents on fracture surfaces [18,19,20,22]. Application of the same targeted therapies to spinal fusion repairs, however, was uncertain because minimal hydroxyapatite is exposed during preparation of adjacent vertebrae for spinal fusion, and the vasculature required for drug delivery was generally disrupted during surgery. Thus, there still remained significant questions about whether the scaffolds commonly used for promoting spinal fusions would recruit an acidic oligopeptide with its attached bone anabolic agent and thereby lead to accelerated mineralization. In the studies above, we demonstrated that even where the scaffolds were composed entirely of collagen, we were able to concentrate systemically administered acidic peptide-targeted payloads within the scaffold matrix. Although such collagen scaffolds were devoid of hydroxyapatite at the time of implantation, we speculate that they rapidly nucleate hydroxyapatite and thereby create a high-affinity surface for recruitment of Ab_46_-D-Glu_20._ With the concurrent vascularization of the scaffold [36], conditions are rapidly generated for selective delivery and accumulation of Ab_46_-D-Glu_20_ at the fusion site. While this postulated chronology requires a brief delay between the surgical procedure and subsequent systemic administration of Ab_46_-D-Glu_20_, the delay can be readily compensated for by soaking the sponge in Ab_46_-D-Glu_20_ prior to implantation. Taken together, this strategy should allow for broad applicability of acidic oligopeptides in targeting scaffolds following skeletal surgeries.

A second question that logically arose is whether abaloparatide constituted the ideal payload for accelerating the fusion process. Although the PTH1R agonist, abaloparatide, has been well established as a bone anabolic agent, it is only clinically approved for treating osteoporosis. Most of abaloparatide’s applications have been limited to promoting systemic changes throughout the skeleton to reduce fracture risk. Because these moderate skeletal changes occur gradually over a period of months, the concern arose that abaloparatide may not prove effective in mediating fusions in the timeframe required for rapid stabilization of a moving joint. In fact, past efforts to use nontargeted abaloparatide for accelerating fracture repair have been unsuccessful due to the lack of efficacy at systemically tolerable doses. However, with our targeting system’s ability to sustain a localized signal at a focal point in the body, the anabolic signal from abaloparatide can be concentrated at and restricted at the site of skeletal damage. This allows abaloparatide to overcome issues with safety and potency. Indeed, we demonstrated above that Ab_46_-D-Glu_20_ could be a potent alternative to BMP2. Additionally, abaloparatide is a smaller anabolic with a history of systemic delivery composed of just 34 amino acids, which is roughly one-tenth of the size of BMP2, making it an ideal candidate for targeted delivery. Abaloparatide is also an ideal payload candidate as it has been shown to promote trabecular bone growth both preclinically [37] and clinically [38]. The vertebral bodies are composed mainly of trabecular bone and play an important role in the overall strength of the spine [39]. Targeted abaloparatide resulted in faster mineralization of the scaffolds. This osteoanabolic response is likely a result of the increased number and activity of osteoblasts recruited to the scaffolds by abaloparatide engagement with PTH1R [29,30] in addition to its angiogenic signals through engagement with PTH1R on endothelial cells, which helped to promote vascularization of the scaffolds. This localized anabolic effect due to the restricted distribution from the targeting moiety not only helps abaloparatide overcome its potency issues but also solves the major challenges that BMP2 has of leakage into the surrounding tissue, which leads to serious edema and ectopic mineralization.

Based on the emerging hypothesis that all an acidic peptide-targeted bone anabolic might need to promote enhanced ossification is granules of hydroxyapatite, we speculate that similar conjugation might be exploited to facilitate healing of other surgical indications involving the use of scaffolds. Such a system could be used to reduce the time for ossification in the lengthy repairs of indications like osteogenesis distraction used to lengthen limbs, and maxillofacial reconstructions, which can impair the ability to eat food for months. Additionally, one could envision applications in supporting the repair of large voids left in bone filled with scaffolds, whether this be due to traumatic injury, debridement in osteomyelitis, or bone-based cancers. This ability to noninvasively administer therapeutics allows for broad integration into numerous disease states with extended repair windows that rely on ossification of a scaffold. Additionally, the ability to administer the therapeutic ad libitum allows the ability to extend the impact of any anabolic over the current standard single topical bolus system.

Although the fusion process in rats treated with BMP2 was found to proceed slower than with Ab_46_-D-Glu_20_, it is conceivable that it nevertheless would have reached the same complete healing endpoint. The question therefore arises as to whether speed is important. Current spinal fusion treatment protocols have lengthy windows of repair with months of severe restrictions on mobility. Not only do these mobility restrictions impact quality of life, but in the primarily older population who undergo spinal fusion surgeries, they can result in increased cardiovascular risks that can be life-threatening. Speed of ossification also reduces the chance of pseudoarthrosis, or hardware failure, thus playing a role in the overall success and thereby the reduction of future revision surgeries.

This bone targeting system and, more specifically, Ab_46_-D-Glu_20_ have broad potential benefits for spinal fusion patients in reducing side effects, improving spinal fusion success rates and reducing surgical recovery time, and, as such, should be developed for clinical translation. With our targeting platform, we can administer drugs multiple times and continue to stimulate bone growth in the fusion region for a longer and faster time with less-invasive drug administration, thus potentially allowing people to regain their post-surgery mobility more quickly and reduce their immobility-associated comorbidities. This acceleration of healing could dramatically reduce the burden of disease and reduce the need for revisionary surgeries [5]. The noninvasive method of this drug’s administration, and its universal application to existing scaffolds, should allow for the ease of its augmentation into current clinical practices. This is a promising potential application of this technology, and future studies should be conducted to further evaluate its potential and see if these preclinical findings translate into clinical improvements for patients recovering from spinal fusions.

In conclusion, we demonstrated that our bone-targeting platform allows for specific localization to spinal fusions. We demonstrated that an osteoporosis drug, when targeted by the system, was found to accelerate both the mineralization rate and the overall fusion rate in rats. It is a system that works with many of the common scaffolds employed in current clinical practice and could be used in conjunction with many of the current surgical practices. It allows for noninvasive administration of drugs such that you can continue to administer the drug to maintain therapeutic levels long after the surgery. It specifically retains the drug just at the site of the fusion so that off-target effects are minimized. This is a promising preclinical finding, and it merits continual clinical development of this therapeutic to see if these findings can be replicated in humans. If validated in humans, it could help neurosurgeons and orthopedic surgeons in treating their patients and helping them to return to functioning sooner. 

## Figures and Tables

**Figure 1 biomedicines-12-00612-f001:**
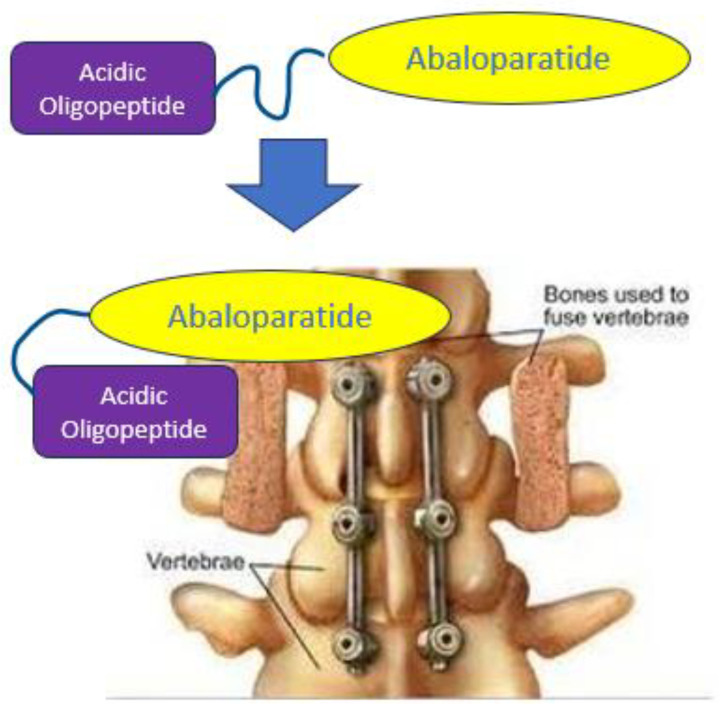
Molecular overview of spinal fusion targeting therapeutic. We synthesized fracture-targeted anabolic agents by synthesizing abaloparatide factors (yellow) in tandem with a spacer (blue) and a hydroxyapatite-binding acid oligopeptide (purple).

**Figure 2 biomedicines-12-00612-f002:**
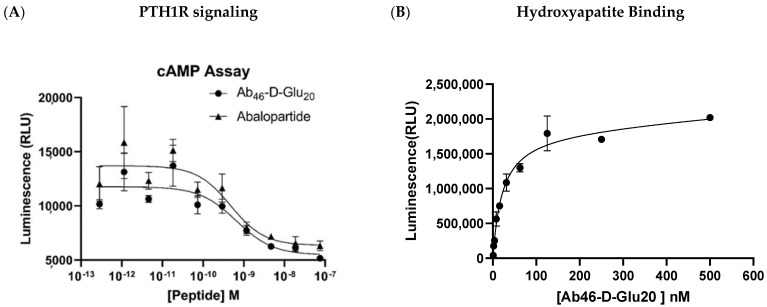
Demonstration of targeted abaloparatide’s retained cellular activity and affinity to hydroxyapatite. (**A**) The activity of targeted abaloparatide relative to free abaloparatide activity was assessed in a cAMP Glo assay on UMR 106 cells treated with drug for 1 h EC_50_ 0.48 nM and 0.66 nM, respectively. The addition of targeting moiety had minimal effects on the engagement of PTH1R. (**B**) In vitro assessment of binding affinity of acidic oligopeptide targeted abaloparatide to hydroxyapatite. Even with the addition of linkers and large abaloparatide payloads, the acidic oligopeptides had a high affinity interaction with hydroxyapatite. The KD binding constant was calculated to be 27 nM. Adapted with permission from Low et al., 2022 [20].

**Figure 3 biomedicines-12-00612-f003:**
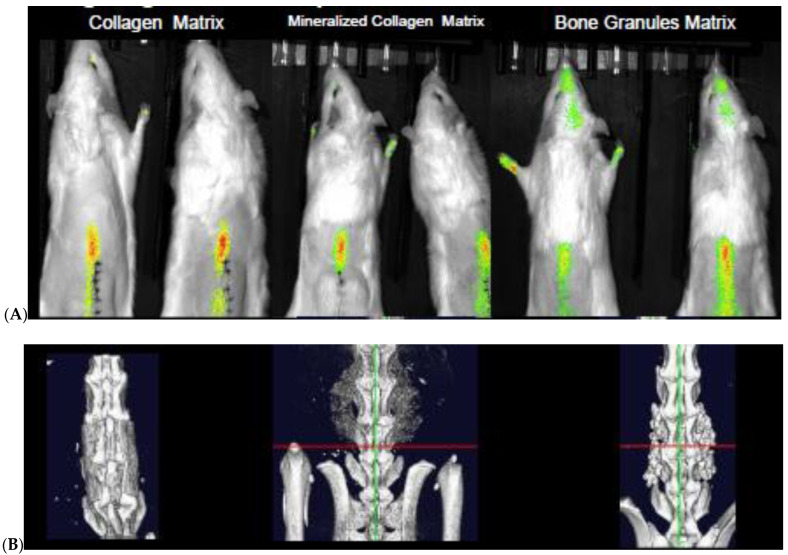
Biodistribution of bone targeting system in rats with L4–L5 posterolateral lumbar fusion with the three most common materials for spinal fusion. To evaluate the ability to localize therapeutics to spinal fusions, female Sprague Dawley rats (*n* = 3) underwent an L4–L5 posterolateral lumbar spinal fusion using collagen sponge and inorganic bone granules wrapped in collagen or a mineralized collagen scaffold soaked in 10 µg BMP 2. Using a low-speed bur, the L4 and L5 transverse processes cortical bone and the facet joint between the two vertebrae were removed, and a drug-soaked collagen sponge was placed on each side of the vertebrae. The scaffold was soaked to saturation in the drugs for 10 min. They were imaged in a µCT (Perkin Elmer) longitudinally every week for 8 weeks and injected with a near-infrared dye (S0456 acidic oligopeptide conjugate and imaged 24 h later with a spectral imaging system) for 1 s with excitation at 745 nM and emission collected at 810 nM. (**A**) Localization of acidic oligopeptide targeted near-infrared dye in a rat with collagen scaffold 2 weeks after surgery, 24 h post-injection. Note localization seen on paws and head are due to grooming with paws contaminated by dye urinated out. (**B**) 3D reconstruction of three different types of spinal fusion scaffolds that were evaluated 4 weeks post-surgery. Red and green lines are cursers from the Perkin Elmer 3D reconstruction software.

**Figure 4 biomedicines-12-00612-f004:**
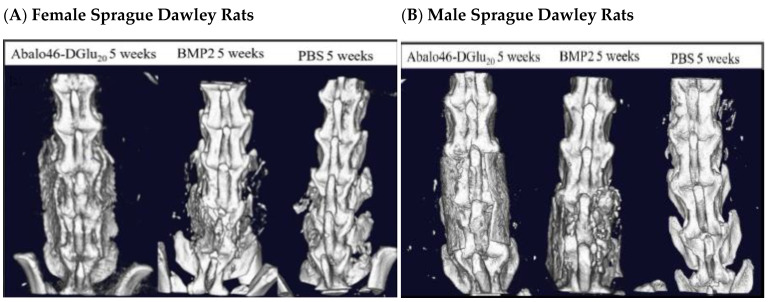
Ab_46_-D Glu_20_ Efficacy study in both sexes using collagen sponges scaffolds. The Sprague Dawley rats (*n* = 10) study was performed using the same surgical procedure as before with a 5 × 75 mm collagen sponge (RCM 6 Ace Surgical supplies). A total of 60 rats were randomly assigned to one of three treatments: phosphate-buffered saline vehicle control, 10 µg BMP 2, or 38 nmol/kg twice a week of acidic oligopeptide-targeted abaloparatide (Abalo 46 D-Glu 20. The initial dose of all drugs was administered on the sponge, and then vehicle control and Abalo 46 D-Glu 20 were administered subcutaneously twice a week for 8 weeks. Images here are median-density images for each group after 5 weeks of treatment. (**A**) represents female rats and (**B**) represents male rats.

**Figure 5 biomedicines-12-00612-f005:**
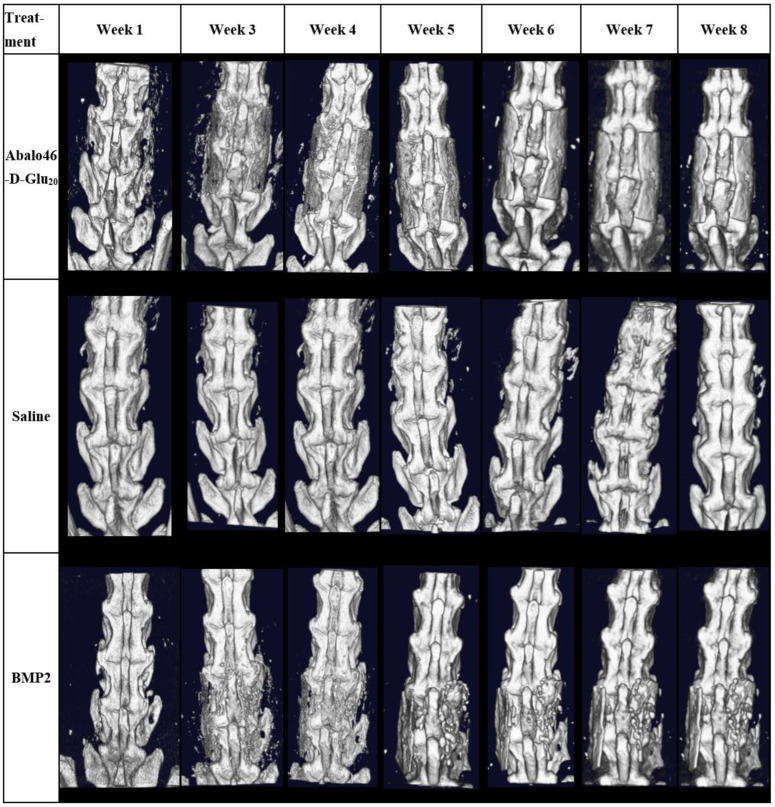
The Sprague Dawley rats (*n* = 10) study was performed using the same surgical procedure as before with a 5 × 75 mm collagen sponge (RCM 6 Ace Surgical supplies). A total of 60 rats were randomly assigned to one of three treatments: phosphate-buffered saline vehicle control, 10 µg BMP 2, or 38 nmol/kg twice a week of acidic oligopeptide-targeted abaloparatide. Images here are median-density images for each group longitudinally in the male rat group. The figures above demonstrate that targeted abaloparatide increased the mineralization of the scaffolds more than BMP2 or saline and achieved this mineralization at earlier time points.

**Figure 6 biomedicines-12-00612-f006:**
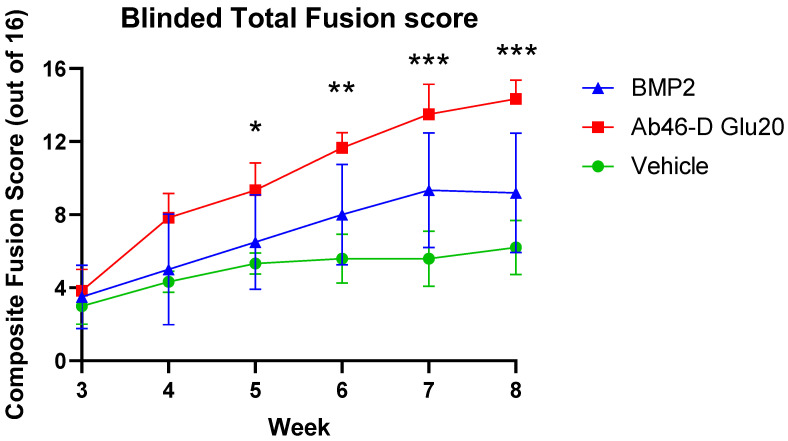
In vivo fusion healing efficacy of targeted abaloparatide on male Sprague Dawley rats (*n* = 10) over 8 weeks after bilateral posterolateral lumbar fusion of L4–L5 analysis. Total fusion scores were assigned as 0–4 for osteointegration, density, and bridging of both bilateral spinal fusions in micro CT scans of the rats. Scores were assigned by two individuals blinded to the treatment reviewing randomized samples. Total score is the sum of all 4 scores combined. Total fusion scores are represented as mean ± SEM (n = 10). Comparisons between the groups were performed using 2-way repeated measures analysis of variance (ANOVA), whereas differences between means were inspected with Dunnett’s multiple comparison post hoc tests in GraphPad Prism version 9. An adjusted *p* value < 0.05 was considered statistically significant and is denoted by *, *p* value < 0.01 was denoted by **, and a *p* value < 0.001 was denoted by ***.

**Figure 7 biomedicines-12-00612-f007:**
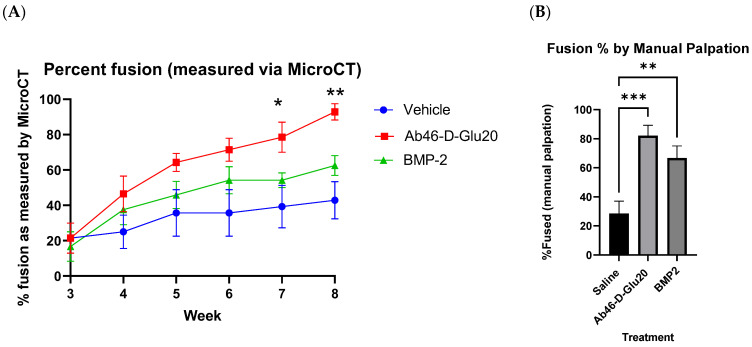
(**A**) In vivo fusion healing efficacy of targeted abaloparatide on male Sprague Dawley rats (10 over 8 weeks after bilateral posterolateral lumbar fusion of L4–L5). Fusion is assessed by completed osteointegration as seen by micro CT. This was assessed in a randomized fashion by 2 individuals blinded to the treatment groups. (**B**) In vivo fusion healing efficacy of targeted abaloparatide on male Sprague Dawley rats (10 after 8 weeks after bilateral posterolateral lumbar fusion of L4–L5 analysis). The lumbar region was excised postmortem and manually palpated and evaluated for fusion between L4 and L5. Percent fusion and fusion % by manual palpation are represented as mean ± SEM *(n* = 10). Comparisons between the groups were performed using 2-way repeated measures analysis of variance (ANOVA), whereas differences between means were inspected with Dunnett’s multiple comparison post hoc tests. An adjusted *p* value < 0.05 was considered statistically significant and is denoted by *, *p* value < 0.01 was denoted by **, and a *p* value < 0.001 was denoted by ***.

## Data Availability

The raw data supporting the conclusions of this article will be made available by the authors on request.

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
