# Peer review of "Targeted Delivery of Abaloparatide to Spinal Fusion Site Accelerates Fusion Process in Rats"

_biomedicines, 2024, doi:10.3390/biomedicines12030612_

Round 1
Reviewer 1 Report
Comments and Suggestions for Authors
An interesting project on a poorly described topic.
The article concerns the use of Abaloparatide for Accelerate Fusion Process
My comments:
TITLE:
The title must include information that it is an animal model - rats.
ABSTRACT:
Please add specific and detailed test results, divided into groups and providing probability values ​​(p < 0.05 or p > 0.05)
Introduction
Please increase the literature review related to the topic.
Please add a clear research hypothesis.
Please clearly state the objectives of the study.
Materials and methods
Well and clearly written.
Methods and study group presented in detail and well.
Results
Well and clearly written.
Good quality and clear Tables and figures.
Results well presented.
Discussion:
A broad review of the literature.
A section with work limitations is necessary. It should be mentioned that this is research on an animal model - rats. These are preliminary conclusions and it is advisable to conduct tests on humans in the future.
Please add a paragraph with conclusions at the end.
Please provide some clear conclusions for orthopedists and neurosurgeons.
Author Response
Title: title has been modifired to include "in rats"
Abstracts: The groups discussed in the abstract were both significant and P<0.05 has been added to clarify that the findings fo those groups were significant.
Introduction:
The second paragraph of the introduction has been rewritten to more clearly state the study objective and the research hypothesis.
The first and second paragraph have had more background added to cover more of the recent updates in the field and an additional 10 citations have been included.
The discussion has been revised to clearly indicated that this work was done in rats and that it would be advisabel to do the owrk in humans int he future. The limitations of the study have been more clearly outlined in the revised draft.
A conclusion paragraph has been added to clearly state the findings and implications for neuroseurgeons and othreopedic surgeons.
Reviewer 2 Report
Comments and Suggestions for Authors
This topic is very interesting. The paper is well written, but it needs some, but important, revisions. Look at these:
- It is not clear in the introduction section what is the purpose of this paper. Improve this part.
- "A second question that logically arose is whether abaloparatide constituted the ideal payload for accelerating the fusion process. Although the PTH1R agonist, abaloparatide, has been well established as a bone anabolic agent, it is only approved for osteoporosis, with most of" This part is not clear. What do the authors mean? Revise this part.
- In the discussion section the authors must discuss more about the role of trabecular bone structure. Look at these 2 very important papers: -- doi: 10.1055/a-1962-0181 -- doi: 10.1016/j.jbiomech.2023.111670
- the methods section should be moved before the results
- "This is a promising potential application of this technology and future studies should be done to future evaluate its potential and see if these preclinical findings translate into clinical improvements for patients recovering from spinal fusions." Look at this one: -- doi: 10.3171/2022.6.SPINE22454
- I think it is important to add a conclusion section, reporting the most important findings of this paper.
Overall a good paper.
Comments on the Quality of English LanguageMinor editing of English language required
Author Response
Introduction
The introduction has been rewritten to include a clearer statement of research objective and hypothesis.
Discussion
The paragraph in the discussion regarding the choice of abaloparatide has been revised to add clarity.
Additional information regarding the role of trabecular bone has been added to the discussion.
The methods section’s location is accordance with the published guidelines for authors for this journal.
A conclusion paragraph has been added.
Round 2
Reviewer 1 Report
Comments and Suggestions for Authors
The authors made all suggested changes. Manuscript acceptable.
Reviewer 2 Report
Comments and Suggestions for Authors
Good